# Are We on Course Yet? Functional Behavior Assessment and Behavior Intervention Plan Technical Adequacy in Schools

**DOI:** 10.3390/bs14060466

**Published:** 2024-05-30

**Authors:** Rose Iovannone, Tobey Duble Moore, Jeffrey M. Williams, Sindy Sanchez, Nycole Kauk

**Affiliations:** 1Department of Child and Family Studies, University of South Florida, Tampa, FL 33612, USA; jwilliams32@usf.edu (J.M.W.); sindy.cbcaba@gmail.com (S.S.); nwilloug@usf.edu (N.K.); 2Department of Educational Psychology, Neag School of Education, University of Connecticut, Storrs, CT 06269, USA; tobey.duble@uconn.edu

**Keywords:** functional behavior assessment, behavior intervention plans, technical adequacy

## Abstract

It has been more than two decades since the Individuals with Disabilities Education Act (IDEA; 1997) included language about the use of functional behavior assessments (FBAs) and behavior intervention plans (BIPs) to address the challenging behaviors of students with disabilities in schools. It has been more than ten years since three technical adequacy studies were published that evaluated school-based FBAs’ and BIPs’ inclusion of essential components and found them to be significantly lacking. The aims of this study were to expand upon the previous research by (a) establishing the psychometric properties of the FBA/BIP Technical Adequacy Evaluation Tool (TATE), (b) evaluating the technical adequacy of 135 completed FBAs and 129 BIPs from 13 school districts across a single state, and (c) comparing the findings to previous studies. The results showed that (a) the TATE has moderate but acceptable internal consistency, excellent inter-rater reliability, and good content validity, (b) the technical adequacy scores of the evaluated products ranged between 40% and 50% of the total components, and (c) most of the BIPs had similar flaws to those seen in the previous research; however, improvement was noted in the FBA components. The implications for practice and suggestions for future research are discussed.

## 1. Introduction

Over 50 years of research have cemented the functional behavior assessment (FBA) process as the gold standard for addressing challenging behavior and developing effective interventions [1,2,3]. The goals of FBA are to identify the relation between an individual’s challenging behavior and environmental events and the function or outcome obtained by performing the challenging behavior. This relation between the environmental context and the presentation of challenging behavior is typically summarized through a hypothesis statement that drives the selection and implementation of hypothesis-linked interventions to include in a behavior intervention plan (BIP) [3,4]. These hypothesis-linked interventions help reduce challenging behaviors by directly modifying antecedent events that predict behavioral occurrence and help establish new behavioral repertoires that allow students to access contingencies (including the function) by performing appropriate replacement behaviors [5]. 

The U.S. Individuals with Disabilities Education Act (IDEA) in 1997, subsequent reauthorization in 2004, and amendments in 2015 mandate the use of FBAs and BIPs in school settings for students with disabilities when their challenging behavior results in a change of educational placement and the behavior is determined to be a manifestation of their disability. Although regulations do not mandate proactively using FBAs to address challenging behavior, many schools conduct FBAs to guide the development of behavioral support for students with or at risk of disabilities [6]. Unfortunately, FBAs/BIPs in schools have not always been perceived and implemented as intended [7,8]. This may be due to inconsistent or ambiguous descriptions of the components and actions required for an adequate FBA/BIP [9,10] and the skills necessary to conduct them [11,12,13]. Additionally, there may be an unwillingness to conduct FBAs/BIPs as intended due to educators’ perceptions of the number of resources and time required [14]. Systems variables can also contribute to the poor quality of FBAs/BIPs, specifically how to modify and standardize procedures that were initially applied in clinical settings to have a contextual fit within typical and often complex school settings but still retain the core underlying principles that make FBAs/BIPs efficacious [8]. While students who typically require FBAs and BIPs may only account for an estimated 5% of the school population, the intensity of the challenging behavior exhibited by these students often consumes many available resources [15]. Therefore, it is critical for the FBA/BIP process to be carried out proficiently, as ineffective interventions derived from poorly completed assessments expend valuable time and effort [5,8].

Despite the fact that policy has required FBAs/BIPs in schools since the passing of the IDEA in 1997, very few studies have evaluated the quality of FBAs/BIPs conducted in school settings. Between 1997 and 2022, three studies were published that evaluated the technical adequacy of FBAs/BIPs completed by typical school personnel. While these studies are seminal to the work evaluating the status of FBAs/BIPs in schools, there remains a dearth of research in this area. The first, conducted by Van Acker et al. [16], created an 11-item checklist using a 6-point Likert scale to evaluate critical components of 71 FBAs and BIPs developed in 70 schools from 21 districts across 1 state. The results indicated that most of the FBAs/BIPs were missing essential components. The common flaws observed were poorly defined target behaviors, lack of confirmation of the hypothesized function, absence of interventions linked with the hypothesis, and minimal plans for ongoing monitoring of implementation fidelity and evaluation of behavior change. In the second study, Blood and Neel [17] evaluated FBAs and BIPs developed for students with emotional/behavior disorders (EBDs) in one school district. Using a self-developed checklist, the authors examined 15 FBAs for the inclusion of specific components and found only 1 that included a hypothesis. The BIPs primarily consisted of a list of potential positive and negative contingencies following challenging behavior. The authors concluded that the FBAs/BIPs appeared to be completed as a compliance activity rather than a process that would result in the development of effective interventions. They also suggested that the skill level required for FBAs/BIPs may not be easily learned through typical school professional development modes (e.g., one-day workshops). The third study [18] evaluated 320 BIPs completed by 2 separate groups of educators in 1 state. Here, 1 group consisted of 244 plans developed by educators identified by special education directors as having the most behavioral expertise and receiving a 6-hour professional development opportunity focused on BIP development. The second group comprised 76 plans completed by typical educators from 1 large urban school district. The measurement tool used was the Behavior Support Plan Quality Evaluation (BSP-QE) [19], a 12-item checklist with a 3-point Likert scale, ranging from 0 (poor or unmet) to 2 (adequate), that rated the extent to which key BIP components were addressed. The results indicated that 65% of the BIPs from the group identified as having higher expertise had adequate scores compared to 11% from the typical educator group. Examination of the plan components indicated that both groups displayed flaws in specifying behavioral goals and objectives and team communication. Within the group of typical educators, low mean scores were observed in relation to developing antecedents, replacement behaviors, and consequence strategies. The authors concluded that educators may not be sufficiently trained to develop adequate BIPs. This conclusion was further supported by the results showing that even the group with more expertise had 35% of their plans scored as inadequate.

Over a decade has passed since the three technical adequacy studies were published. A new technical adequacy study to determine whether schools have improved the quality of FBAs/BIPs seems needed. A recent study by Hirsch and colleagues [20] evaluated 304 FBA/BIP electronic documents from 1 school district to determine the degree to which the documents met expectations for including FBA/BIP components. Using a self-developed 14-item rubric with a 3-point Likert scale, the items were coded as 0 = not meeting expectations, 1 = approaching expectations, or 2 = meets expectations. The results indicated that approximately 50% of the records did not meet expectations regarding key FBA/BIP components; however, the authors noted an improvement from the previous reviews (e.g., [16,17,18]) in that many FBA components were scored as approaching meeting expectations. Two primary limitations were noted. First, many items could not be coded due to the unavailability of specific information in the electronic files, which forced them to recalibrate their rubric to match what was available electronically. Second, the study only reviewed one school district, limiting the conclusions to be drawn. Thus, the purpose of this study was to conduct a descriptive study, similar to Van Acker’s [16], to re-examine the current technical adequacy of completed FBAs/BIPs from school districts across one state, using a researcher-developed scoring tool and rubric titled the FBA/BIP Technical Adequacy Tool for Evaluation (TATE). Additional purposes included (a) determining the extent to which the evaluation tool was reliable and valid and (b) comparing the current technical adequacy themes from this study with the initial studies conducted by Van Acker, Blood and Neal, and Cook [16,17,18] and determining if adequacy is improving. The research questions in this study were as follows:What are the psychometric properties of the TATE?To what extent do current completed FBAs/BIPs produced by school teams in districts throughout the state include essential components?How do the current technical adequacy outcomes compare with the initial technical adequacy studies conducted in the previous decade?

## 2. Method

### 2.1. Participants and Setting

The FBA/BIP documents were received from local education agencies (LEAs) throughout a southeastern state. A total of 171 FBAs/BIPs, with any personally identifying information (e.g., names of students, educators, schools, birthdates) redacted, were submitted for review from 62 schools in 13 LEAs, representing 19% of the LEAs and 1% of the schools across the state. The documents came from two sources. One set (*n* = 123) was submitted voluntarily by six LEAs to assist with validating a tiered PBIS fidelity measure. The other set (*n* = 48) came from seven LEAs identified by the State Department of Education as having a high number of restraint/seclusion incidents. They submitted FBAs/BIPs developed for students who received a higher amount of restraint/seclusion events to provide technical assistance to the LEAs in improving practices. Both sets of data were collected as part of two state discretionary projects’ technical assistance activities provided to the State Department of Education and LEAs for improving behavioral outcomes for students with or at risk of disabilities. All the documents were submitted either through attachments to emails (doc., docx., or .pdf), faxed, or mailed via UPS to the authors. Of the 171 documents submitted, 36 were eliminated from the coding due to only submitting a BIP without the FBA, leaving 135 records available for scoring. These records were eliminated due to the Technical Adequacy Tool for Evaluation (TATE; measure used and described in next section) needing the FBA to code the BIP. However, if the document included the FBA but omitted the BIP, the FBA was scored while the BIP was coded as “missing”. This situation occurred with six submitted documents. The final number of documents scored was 135 FBAs and 129 BIPs. Of the documents scored, 82 (57%) FBAs and 76 (56%) BIPs were from elementary schools, 15 (%) FBAs and BIPs were from middle schools, 8 FBA/BIPs (6%) were from high schools, 24 FBAs/BIPs (16/17%) were from special education center schools, and the remaining 6 FBA/BIPs (4%) were from those classified as multiple grade schools (i.e., elementary/middle school combo, K-12 school). The majority of the scored documents came from LEAs ranked as large (78/55% FBAs and 77/57% BIPs), with 47 FBAs (33%) and 42 (31%) BIPs coming from very large, and 10 FBAs (6%)/BIPs (8%) from medium (5) and small (5) LEAs. Due to the redactions of personally identifiable information, it is unknown what educator role completed the submitted FBA/BIP documents. Additionally, some district FBA/BIP form templates did not have an option for that information to be included.

### 2.2. Measure

#### FBA/BIP Technical Adequacy Tool for Evaluation (TATE)

To evaluate each FBA/BIP, we used the TATE [21] developed by a university research team that included two doctoral faculty, both board-certified behavior analysts, and one master’s level staff member. We decided to develop a new tool as opposed to using the previously developed tools in the aforementioned studies for two reasons. First, we wanted to have a tool that aligned with what the literature identified as essential features that comprise quality FBAs and BIPs, not just BIPs. Second, we wanted to develop a tool that could be generalized for evaluating any FBA and BIP rather than a tool with items reliant on a specific approach, form template, model, or training curriculum. The TATE items were created by reviewing the behavior analysis literature, existing measures and checklists (e.g., Behavior Support Plan-Quality Evaluation Guide [19], Individual Student Systems Evaluation Tool [22]), and legal decisions related to FBAs/BIPs. The TATE items were worded so that the tool can be used to evaluate any FBA/BIP, regardless of the specific forms and processes used by any school district. The TATE includes a scoring form and a scoring rubric.

The content validity was checked by sending the first draft of the TATE scoring tool and the rubric, which at that time included twenty items, to three national experts who had a minimum of three peer-reviewed articles published on FBAs/BIPs within the previous five years. The experts were asked to review the TATE scoring tool and respond to three questions: (1) Is the component essential? (2) Should the component be combined with another? (3) Is the item written clearly? Three questions were asked concerning the rubric: (1) Do the descriptors differentiate the scores? (2) Are the descriptors written clearly? (3) Are the scoring examples adequate? Suggestions for edits and refinements were offered for any “no” responses. After the expert review, edits were made to increase the clarity. Two items in the BIP section that evaluated the adequacy of a data-monitoring plan for both the challenging and the replacement behavior were combined, and one FBA item that evaluated the inclusion of a reinforcer assessment was eliminated. This resulted in the final version of the TATE consisting of 18 items, 9 evaluating the FBA and 9 evaluating the BIP. Each item is rated using a 3-point Likert scale, with a 0 indicating the absence or minimal presence of the component, a 1 indicating partial presence, and a 2 indicating adequate presence. The total raw score obtained on the TATE is 36, with a total raw score of 18 for the FBA and the BIP sections. A rubric accompanies the TATE, defining the scores for each item and providing examples of responses. For example, item 1, which asks about the sources of FBA data, describes a “1” score as follows: “Vague indication that input was collected from more than one person/source; details missing; and provides an example as follows: Checklist or list of names of people who participated in the FBA but no indication of what data or sources were attributed to them.” The nine FBA components include (a) sources of FBA data, (b) operational definition of the target behavior, (c) baseline data of the target behavior, (d) setting events, (e) antecedent events related to challenging behavior, (f) antecedent events related to appropriate behavior, (g) responses following challenging behavior, (h) hypothesis statement, and (i) valid function. The nine BIP components include (a) timeline between FBA and BIP, (b) inclusion of hypothesis, (c) antecedent strategy, (d) teach/replacement behavior strategy, (e) reinforcement of replacement behavior strategy, (f) response to target behavior strategy, (g) safety plan, (h) progress monitoring plan, and (i) implementation fidelity plan. 

The convergent validity (i.e., comparing the TATE with another similar instrument) was not checked for several reasons. First, out of the three studies cited, only the BSPQE [19] was available as a tool for scoring with the psychometric properties recorded. The other two evaluation tools used by Van Acker [16] and Blood [17] were not in a published format and did not have reliability or validity reported. Second, upon inspecting and comparing the BSPQE with the TATE, they are dissimilar in that the BSPQE only scores the BIP while the TATE scores both the FBA and BIP. Third, the BSPQE was designed for use with specific FBA/BIP training provided by a state project. For example, the scoring rubric of the BSPQE would provide instructions to review particular lines or sections of the state project FBA/BIP template form to determine the item’s rating. These issues resulted in the authors deciding not to conduct additional validity evaluations.

### 2.3. Procedures

#### 2.3.1. Training of Raters

Before scoring the FBAs and BIPs, the primary author trained six advanced graduate students in applied behavior analysis and school psychology to use the TATE. The training included a one-hour didactic presentation in which each item’s rationale was explained with examples of full, partial, and non-examples for items. The raters then used the TATE to score practice FBAs/BIPs that were not part of the dataset for this research study and were provided with feedback on their performance to improve their ability to score products reliably. Training continued until all the raters scored two practice plans with 80% inter-rater reliability (IRR) with the first author. After achieving this criterion, each rater was provided a randomized subset of 20–25 FBAs/BIPs to score.

#### 2.3.2. Rating Procedures

Three rating dyads were formed, with two raters in each dyad. An Excel database was created, with each document having a unique research identification number and cells to record the variables. The file was saved in a shared university drive (i.e., Box). 

A total of 20% of the FBAs and BIPs assigned to each rater were scored independently by the other rater in the dyad, with 10% of the 20% being scored by a third rater (first author). The IRR was calculated for each of the 18 components, and the overall IRR was calculated as (total agreements)/(total agreements + total disagreements) × 100. The IRR scores ranged from 75% to 97%, with an overall mean IRR of 91% across the six raters. Ratings in disagreement were discussed and resolved, with the final agreed-upon score used in the analysis.

### 2.4. Data Analytic Strategy

As described previously, when a BIP was submitted without the FBA, the documents were not coded and were identified as missing data. If the FBA was submitted without a BIP, the FBA was scored but the BIP was coded as missing data. Descriptive statistics (means) were used to describe the technical adequacy of the evaluated FBAs/BIPs. Item ratings were treated as ordinal-level variables. The FBA and BIP total scores used numerical means of the sums. After coding was completed and entered into Excel spreadsheets, data were imported into SAS v9.4 for statistical analysis. 

## 3. Results

### 3.1. RQ1: Reliability and Validity of TATE

#### 3.1.1. Inter-Rater Reliability

The inter-rater reliability of the TATE was examined at both the total score and individual item levels. Intraclass correlation coefficients were calculated for the total scores, yielding 0.92 for the FBA total score, 0.93 for the BIP total score, and 0.94 for the overall total score, indicating excellent agreement across the raters. Due to the ordinal nature of the TATE item scoring (i.e., three categories), weighted kappa coefficients were calculated for the item-level analyses as they take into consideration the level of disagreement (see Table 1). The kappa coefficients ranged from −0.03 to 0.97, but if the lowest coefficient (−0.03) was ignored, the minimum would be 0.57, with half of the items having a coefficient of 0.80 or above. 

#### 3.1.2. Internal Consistency Reliability

Due to the ordinal nature of the data, the ordinal alpha was used to evaluate the internal consistency of the total scores [23]. Convergence errors were encountered for the overall total scores, but the alphas for the FBA and BIP total scores were 0.66 and 0.62, respectively.

#### 3.1.3. Validity

Finally, to examine the validity of the TATE, an ANOVA was conducted to compare the scores between the two sources of the products (i.e., those submitted for validating a tiered measure and those submitted by the State Department of Education for high rates of restraint and seclusion) scored with the TATE. Given the highly unequal group sizes (N = 47 and 88 for the restraint/seclusion set and the tiered measure validation set, respectively), we modeled separate residual variances for the groups in SAS Proc MIXED [24]. The overall test of the group differences for the FBA total score was not significant (*F*[1, 133] = 2.69, *p* = 0.104), but that for the BIP was significant (*F*[1, 127] = 9.3, *p* = 0.003). Examination of the group means was consistent with the hypotheses. The group from the tiered measure validation set had higher scores on the FBA and BIP than those with high restraint/seclusion incidents (9.55 vs. 8.81 and 8.02 vs. 6.55, respectively).

### 3.2. RQ2: Current FBA/BIP Descriptive Statistics

In answering research question two, Table 2 provides the overall mean raw scores for the FBA and BIP, while Table 3 provides the mean scores for each item. As can be observed, the overall mean raw scores of 9.30 for the FBA and 7.49 for the BIP indicate that the submitted FBAs/BIPs from the school districts were lacking in many of the essential components. The item mean scores showed that the FBA components receiving the highest ratings were (a) having multiple sources of input/data for the FBA (1.49), (b) clearly identifying and defining the target behavior (1.49), and (c) having a hypothesis statement with a valid function (1.47). The FBA item mean scores receiving the lowest ratings were (a) describing setting events and their relation to the occurrence of the target behavior (0.34), (b) identifying antecedent events present in the absence of target behavior (0.49), and (c) identifying responses/consequences following the target behavior (0.78). The mean scores for the BIP section showed that all the items except for the first two were between 0 and 1. The highest-rated items in the BIP section included (a) timeline between the FBA and BIP (1.51), and (b) hypothesis included or referenced in the BIP (1.33). The lowest-rated BIP items included (a) describing how implementation fidelity will be measured (0.08), (b) describing an intervention that changes responses to target behavior (0.48), and (c) describing a hypothesis-linked intervention to reinforce the replacement behavior (0.70).

Table 4 provides the relative frequency distributions for each TATE item’s ratings (0, 1, 2). The three items receiving more ratings of 2 included (a) item 10-Timeline Between FBA and BIP (98/74.8%), (b) item 1-Multiple Sources of FBA Data (79/58.1%), and (c) item 2-Operational Definition of the Target Behavior (67/49.3%). The items receiving more ratings of 0 include (a) item 18-Plan for Measuring Implementation Fidelity (121/93.1%); (b) item 2-Describing Setting Events (96/70.6%), and (c) item 6-Describing Antecedent Events in the Absence of the Target Behavior (90/66.2%).

Table 5 compares the mean total raw scores across the sources of the documents (i.e., validating tiered fidelity measure or restraint/seclusion identification) and across the school grade levels. The documents submitted by LEAs for validity testing had higher mean scores for the FBAs and BIPs than those submitted for students identified as receiving restraint/seclusion practices. When comparing the FBA mean scores by the school grade level, the high school documents had the highest mean scores (11.3), followed by the elementary schools (9.7). The documents from special education self-contained schools had the lowest FBA mean scores (8.0), while the middle schools had the lowest BIP mean score (6.3). The highest mean BIP scores came from the high and combined grade schools (8.3 for both). The middle school documents had the lowest BIP mean score (6.3).

### 3.3. RQ3: Comparison of Current Themes to Past Research

In answering research question three, which examined whether the results of this study show the same patterns of inadequacy as earlier studies, Table 6 lists the TATE components that most closely match the FBA/BIP components measured in other studies and found to be inadequate in at least one other study. Items in bold font were identified as inadequate in two or more studies. Items identified as inadequate in each study are indicated with an “x”. The results show continuing corroboration of inadequacy in terms of several BIP components. Specifically, interventions that describe teaching replacement or alternate appropriate behaviors, using the hypothesized function as a reinforcement for the performance of the replacement behavior, and changing responses to target behaviors so that they are no longer inadvertently reinforced continue to be areas of weakness. The TATE component results shared with at least one other study are italicized. Antecedent or prevention strategies and the inclusion of a progress monitoring plan were shared by the TATE and Blood and Neal’s [17] and Van Acker et al.’s [16] studies (respectively). No similarity of themes was seen between the TATE FBA items and previous studies.

## 4. Discussion

This study explored the technical adequacy of completed FBAs and BIPs in one state. Our specific goals were threefold. First, we wanted to examine the psychometric properties of the TATE, the tool used for evaluating the technical adequacy of the completed FBAs and BIPs. Second, we wanted to explore the technical adequacy of 135 FBAs and 129 BIPs completed by school-based educators across a single state. Third, we wanted to explore whether the technical adequacy outcomes of the FBAs and BIPs evaluated in the current study had improved compared to three similar studies conducted almost two decades earlier.

To accomplish this, we first evaluated the inter-rater agreement when using the TATE. Our findings indicated that the TATE is a reliable tool for measuring the technical adequacy of FBAs/BIPs, as demonstrated by the high inter-rater reliability across the total FBA and BIP scores and most of the individual items. Item 18, which evaluated how the implementation fidelity of the behavior intervention plan would be measured, was an outlier due to the very low incidence of non-zero responses by raters (92% zeroes). The content validity of the TATE was established through feedback from three expert reviewers.

Regarding our second research question, the FBAs and BIPs assessed in this study showed low overall rates for technical adequacy. The raw scores on the TATE were higher for the BIP items than for the items in FBAs. There may be several reasons for this difference. First, although competencies in ABA principles are essential for conducting FBAs, the components are more manualized and trainable than those required to determine appropriate interventions in a BIP. For example, teaching educators to write operational definitions of target behaviors is a relatively straightforward skill. Educators can be provided with critical elements of an operational definition, such as “observable” and “measurable”, as well as examples and non-examples. When collecting direct or indirect data, various tools exist to guide this process (e.g., the FACTS), which include a competing behavior pathway and provide structured guidance for writing a hypothesis or summary statement [25]. Comparatively, the skills needed to determine appropriate BIP interventions from a corresponding FBA are more complex and require more individualization based on the student and setting. Despite these differences, various studies have demonstrated positive outcomes in training the skills needed to produce both an FBA and BIP, including knowledge gain and fidelity of implementation [11,13,26]. It is also possible that while successful training outcomes in both areas are possible, especially in a research setting, in practice, actual skill levels for conducting FBAs are higher than those for BIPs. It is possible that the mean score for “Implementation Fidelity Plan”, which was very low (0.08), may have lowered the overall mean for the BIP scores. Prior research has shown that, in practice, behavior plans are often implemented without fidelity [18,27], which can have a negative effect on student behavioral outcomes [28].

There were some relative areas of strength in the examined FBAs/BIPs, typically within the early and less technical items from the assessment/intervention process. For the FBAs within this dataset, the relative strengths included having multiple sources of input and data for the FBAs and clearly identifying and defining the target behavior, which may indicate that structures and expectations for conducting FBAs have improved to ensure that various individuals are included and multiple data sources are utilized to develop a hypothesis. Interestingly, another relative strength was having a hypothesis statement with a valid function, despite many FBAs lacking data on setting events, antecedents, and consequences—the data typically used to create a function-based hypothesis. Again, this may be because creating a hypothesis statement is a more manualized process that is often outlined in various FBA assessment tools. In contrast, the process of data collection is more open-ended. In addition, this specific item is included to ensure that functions that are unobservable or non-measurable constructs, such as “control” or “power”, were not identified as the maintaining consequence of challenging behavior [29].

Regarding the BIP scores, it is promising to see that many BIPs are being created in a timely manner following an FBA overall and that most BIPs at least reference the hypothesis statement from the corresponding FBA. The purpose of the FBA is to develop a corresponding BIP, so a relatively quick timeline between the two and the presence of the FBA hypothesis increases the likelihood that function-based interventions are chosen in the BIP.

In relation to research question three, regarding previous studies examining the technical adequacy of FBAs, the overall results obtained in this study continued to corroborate the findings of Van Acker et al. [16], Blood and Neal [17], and Cook et al. [18], particularly regarding ratings of items from BIPs. The most common similarity between this study and past studies is the relatively low-rated scores for (a) identification of a functionally similar replacement or alternate behavior and a teaching strategy that provides the student with an alternate pathway to obtaining the hypothesized function by performing the appropriate behavior, (b) providing reinforcement when the student engages in appropriate behavior, and (c) strategies to refrain from inadvertently continuing to reinforce or maintain the occurrence of target behaviors by providing the same function (i.e., escape, access to attention). These continued patterns are alarming, as students are likely not receiving explicit instruction for appropriate replacement behaviors that serve the same function as the target behavior. Without an appropriate replacement skill, students are less likely to decrease the target problem behaviors [30]. These patterns also indicate a lack of appropriate changes in staff responses to student behaviors. To effectively change a student’s behavior, increasing reinforcement of appropriate or replacement behaviors is necessary while simultaneously decreasing or removing reinforcement of the target inappropriate behavior [30]. Without a strategy to implement these changes, staff may inadvertently continue reinforcing a student’s inappropriate behavior, making student behavior change less likely [31].

Compared to previous studies, the current study shows improvements in the FBA components, with many having mean scores of or above 50% (e.g., mean score of 1.0 or higher). Also, the FBA components receiving the highest mean scores in this study were components identified as concerns in previous studies. These components include using multiple sources of informants or data-collection methods and operational definitions of target behaviors. This may reflect system-level improvements at the state or district level regarding expectations around how an FBA should be conducted (e.g., templates for data collection or task analyzed steps regarding collecting data from various informants.

### 4.1. Limitations and Future Research

One of the limitations of the current study is that the TATE does not establish cut-off scores for technically adequate FBAs/BIPs. There are two reasons for this omission. First, we do not have enough FBAs/BIPs with TATE scores above 50%. The distribution is primarily in the lower two quartiles. Second, we do not have empirical support to guide what level of technical adequacy is necessary for an FBA/BIP to be deemed to be of high quality. We could have arbitrarily set benchmarks for adequate and inadequate plans (e.g., scores of 80% and above are adequate); however, we chose not to do so without substantial data confirming this. The goal when developing the TATE was to have a helpful tool that typical educators could use to identify FBA/BIP professional development needs and set goals for improvement. However, even though we do not have official cut-off scores, it would most likely be safe to say that FBAs/BIPs whose TATE scores are below 70% are inadequate, as suggested by the BSP-QE scoring guide [19], which defines “inadequate” plans as those earning 67% or less. Future research is needed, though, to establish cut-off levels that identify adequate FBAs/BIPs.

Another limitation of this study is that even if we can provide a definitive score that separates adequate from inadequate FBAs/BIPs, we still do not know how that cut-off score impacts student outcomes. Cook et al. [32] conducted a study investigating the relation between technically adequate BIPs and student behavioral outcomes, and the results indicated that FBAs/BIPs that were deemed adequate were associated with improved student outcomes, with BIP implementation fidelity serving as a mediator. If we can convincingly show that technically adequate FBAs/BIPs result in higher implementation fidelity and improved student outcomes compared to inadequate FBAs/BIPs, a potential cut-off score could be developed to qualify a document as technically adequate to produce student outcomes or not. More research is needed to add to Cook and colleagues’ study [32], which will explore and solidify the correlations between FBA/BIP technical adequacy, implementation fidelity, and student outcomes,

A third limitation is that we did not collect and evaluate FBAs/BIPs from 100% of the districts in the state. Thus, we may not be able to generalize the findings to all the districts. We also only looked at the FBAs/BIPs from one state; therefore, we cannot generalize our findings to other states. Although previous research [16,17,18] suggests that the pattern of inadequacy is seen in other states, further research is needed to determine if the result of this study can be generalized to districts and states across the U.S. and to determine the patterns of strengths and weaknesses of FBAs/BIPs nationwide.

A fourth limitation is that we did not conduct a convergent validity assessment of the TATE. Although we had considered doing so with the BSPQE, we ultimately decided that the BSPQE had features that would not allow it to be an appropriate measure of concurrent validity, including only evaluating the BIP rather than both the FBA and BIP and being developed to be used for a specific FBA/BIP training and forms rather than a general evaluation of any FBA/BIP. However, the need to perform additional validity evaluations to ensure that the TATE has external validity is an area for future research, particularly if another instrument exists that examines the quality of FBAs and BIPs.

A fifth limitation is that due to all the personally identifiable information being redacted, as well as some of the FBA/BIP district forms lacking a section that would allow a team to enter the roles, we did not have any information on the represented job titles or disciplines of those who completed or participated in the FBAs/BIPs. Thus, we do not know how many of the FBAs/BIPs were facilitated by a role that suggests expertise in the application of behavioral principles (e.g., behavior analysts, school psychologists), nor the level of training possessed by them. There is minimal federal and state legislation or guidance about who is qualified to conduct an FBA [10]. It is possible that, in the case of facilitating FBAs/BIPs, the skills an educator should possess are more important than the role or job title they hold in a school district. There is some research that demonstrates targeted FBA/BIP training and support for educators can improve their technical adequacy [13,17,32]; however, further research is needed to determine the level of training and skill competency level needed as well as to correlate the adequacy of FBAs/BIPs with training.

### 4.2. Implications for Practice

The results of this study also indicate a continued need for systematic improvements to FBAs and BIPs in schools today and for identifying professional development methods that can provide educators with the necessary skills. There is promising research that supports the idea that staff without formal behavior training can gain the knowledge and skills needed to conduct a technically adequate FBA and create a corresponding BIP [11,13,26]. However, data from these studies demonstrate that a research-to-practice gap still exists between what should be included in a technically adequate FBA and BIP and what is actually being conducted in school settings. Further research is needed to determine what systems are required to support districts in capacity building and staff training to ensure that technically adequate FBAs and BIPs are being developed. The TATE is a tool that can be used to help districts in this process. Regular reviews of the FBA/BIP technical adequacy can guide districts in making training, capacity-building, and ongoing coaching decisions.

### 4.3. Implications for Policy

While the IDEA requires that FBAs be conducted under certain circumstances [33], there is minimal federal or state guidance around who should conduct an FBA/BIP, what level of training is needed, what should be present in a technically adequate FBA/BIP, and how the fidelity of implementation or progress monitoring should be measured [10]. While the Office of Special Education and Rehabilitative Services (OSERS) has provided guidance that individuals must “have the content knowledge to serve children with disabilities” [34] (OSERS, 2022, p. 28), they have not specified what that means for individuals conducting an FBAs or BIPs. Federal policy also does not specify what constitutes a technically adequate FBA/BIP. Research has identified various key components of FBAs and BIPs; however, these are not necessarily reflected in any laws regarding conducting FBAs. State policy is also scant in its guidance in this area [10], which may be why the technical adequacy of FBAs/BIPs is so varied and inadequate. The results of this study indicate a need for increased federal and state guidance and regulations around who can conduct an FBA/BIP and what components are required for it to qualify as technically adequate. There is also a need for increased training and support to expand the capacity of trained educators with the skills required to complete a technically adequate FBA and the corresponding BIP.

### 4.4. Conclusions

It has been more than two decades since a series of research studies explored the technical adequacy of FBAs and BIPs completed by school-based educators and found them to need improvement. Our study updated the research on the technical adequacy of FBAs/BIPs conducted in schools and showed similar results to the previous studies, with some improvements in the FBA components. It provided psychometrics on the TATE, a potential tool educators can use to identify areas of improvement. A clear implication of this study is that more support is needed for schools to improve FBA/BIP technical adequacy. In addition, more research is needed to identify what score establishes an FBA/BIP as being technically adequate. Finally, research is needed to determine the association between technical adequacy and student outcomes.

## Figures and Tables

**Table 1 behavsci-14-00466-t001:** Item inter-rater reliability estimates for the TATE items and total scores.

TATE Item	Correlation Coefficient	* Lower	* Upper
FBA items			
Item 1 Sources of FBA	0.82	0.67	0.97
Item 2 Clearly defined target behavior	0.57	0.35	0.78
Item 3 Baseline data	0.78	0.60	0.91
Item 4 Setting events	0.85	0.71	0.99
Item 5 Antecedents for target behavior	0.86	0.73	1.00
Item 6 Antecedents-absence of target behavior	0.88	0.78	0.98
Item 7 Consequences following target behavior	0.63	0.36	0.90
Item 8 Hypothesis linked to FBA	0.70	0.50	0.90
Item 9 Valid function	0.87	0.73	1.00
BIP items			
Item 10 Timeline between FBA and BIP	0.98	0.94	1.00
Item 11 Hypotheses included on BIP	0.65	0.37	0.92
Item 12 Antecedent strategy described/linked to hypotheses	0.57	0.31	0.82
Item 13 Replacement behavior strategy described/linked to hypotheses	0.80	0.62	0.98
Item 14 Reinforce strategy described/linked to hypothesis	0.68	0.39	0.97
Item 15 Changing response to target behavior	0.73	0.51	0.95
Item 16 Consideration and description of safety plan	0.97	0.93	1.00
Item 17 Progress monitoring plan	0.87	0.73	1.00
Item 18 Implementation fidelity plan	−0.03 ^a^	−0.07	0.01
FBA total	0.92	0.85	0.96
BIP total	0.93	0.86	0.96
Total TATE	0.94	0.88	0.97

Note: Item correlation coefficients are represented as Cohen’s kappa (weighted). Total FBA and BIP scores are represented as intraclass correlations. * Lower and upper limits for 95% confidence interval. ^a^ This one low kappa coefficient is due to the very low incidence of non-zero scores by raters (92% zeroes).

**Table 2 behavsci-14-00466-t002:** Descriptive statistics for the FBA mean raw score and BIP mean raw score.

Component (N)	* Mean	Minimum	Maximum	SD
FBA (135)	9.30	2.00	14.00	2.63
BIP (129)	7.49	1.00	15.00	2.74

* Maximum raw score for the FBA and BIP is 18.

**Table 3 behavsci-14-00466-t003:** Mean TATE item scores.

Item	Mean Score (Range = 0–2)	SD
Functional Behavior Assessment		
1. Sources of FBA input/team	1.49	0.66
2. Clearly defined target behavior	1.49	0.50
3. Baseline data	0.99	0.64
4. Setting events	0.34	0.56
5. Antecedents for target behavior	1.18	0.60
6. Antecedents-absence of target behavior	0.49	0.74
7. Consequences following target behavior	0.78	0.78
8. Hypothesis linked to FBA	1.33	0.84
9. Valid function	1.47	0.66
Behavior Intervention Plan		
10. Timeline between FBA and BIP	1.51	0.85
11. Hypothesis included or referenced on BIP	1.33	0.84
12. Prevention strategy described and linked to hypothesis	0.73	0.62
13. Replacement behavior strategy described and linked to hypothesis	0.85	0.53
14. Reinforce strategy described and linked to hypothesis	0.70	0.57
15. Changing response to target behavior	0.48	0.67
16. Consideration and description of safety plan	0.93	0.91
17. Progress monitoring plan	0.85	0.58
18. Implementation fidelity plan	0.08	0.30

**Table 4 behavsci-14-00466-t004:** Relative frequency distributions of the TATE items.

TATE Items	Score 0		Score 1		Score 2	
	*n*	%	*n*	%	*n*	%
1. FBA—Sources of Data	12	8.8	45	33.1	79	58.1
2. FBA—Operational Definition	0	0	69	50.7	67	49.3
3. FBA—Baseline Data	29	21.3	80	58.8	27	19.9
4. FBA—Setting Events	96	70.6	34	25.0	6	4.4
5. FBA—Antecedent Events TB	14	10.3	84	61.8	38	27.9
6. FBA—Antecedent Events AB	90	66.2	26	19.1	20	14.7
7. FBA—Consequences	59	43.7	47	34.8	29	21.5
8. FBA—Hypothesis Statement	13	9.6	100	74.1	22	16.3
9. FBA—Valid Function	12	8.9	48	35.6	75	55.6
10. BIP—Timeline	31	23.7	2	1.5	98	74.8
11. BIP—Hypothesis Match	31	23.8	25	14.6	74	56.9
12. BIP—Prevent Strategy	47	35.9	72	55.0	12	7.0
13. BIP—Teach Strategy	30	22.9	91	69.5	10	7.6
14. BIP—Reinforce Strategy	46	35.4	77	59.2	7	5.4
15. BIP—Change Responding Strategy	81	62.3	36	27.7	13	10.0
16. BIP—Safety Plan	58	44.6	23	17.7	49	37.7
17. BIP—Progress Monitoring	33	25.4	84	64.6	13	10.0
18. BIP—Implementation Fidelity Plan	121	93.1	8	6.2	1	0.8

**Table 5 behavsci-14-00466-t005:** Comparison of the TATE means between groups.

Group	Mean FBA Raw Score (N)	Mean BIP Raw Score (N)
Sources of Documents		
Tiered PBIS Fidelity Validation	9.6 (88)	8.0 (82)
Restraint/Seclusion	8.8 (47)	6.6 (47)
School Grade Level		
Elementary	9.7 (82)	7.6 (76)
Middle	8.6 (15)	6.3 (27)
High School	11.3 (8)	8.3 (8)
Special Education Self-Contained	8.0 (24)	7.4 (24)
Multiple Grade Combo	8.8 (6)	8.2 (6)

**Table 6 behavsci-14-00466-t006:** Comparison of lower-rated technical adequacy components evaluated across studies.

Component	Van Acker	Blood	Cook * AT TT		Current Study
**FBA**					
Sources of FBA	x	x			
Clearly defined target behavior	x				
Baseline data					x
Setting events					x
Antecedents-target behavior				x	
Consequences following target behavior					x
Hypothesis linked to FBA		x			
BIP					
*Prevention strategy described and linked*		*x*			*x*
**Replacement strategy described and linked**	**x**	**x**		**x**	**x**
**Reinforce strategy described and linked**	**x**	**x**			**x**
**Changing response to target behavior**	**x**			**x**	**x**
Consideration and description of safety plan					x
*Progress monitoring plan*	*x*				*x*

Note: The table only includes the TATE components that were also components included in the evaluations used by the other studies. Bold font indicates components found to be similar across 2 or 3 studies. Italicized font indicates components found to be similar with 1 other study. * AT = Advanced Team; TT = Traditional Team.

## Data Availability

Data are available upon reasonable request from the corresponding author.

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
