# Peer review of "Are We on Course Yet? Functional Behavior Assessment and Behavior Intervention Plan Technical Adequacy in Schools"

_behavsci, 2024, doi:10.3390/bs14060466_

Round 1

Reviewer 1 Report

Comments and Suggestions for Authors

I'm glad to review the manuscript and I believe it has the potential to bring big influence in the field on the implementation of FBA/BIP in school settings. Please see the attachment for the detailed feedback.

Reviewer 2 Report

Comments and Suggestions for Authors

Thank you for the opportunity to review the current paper, "Are We On Course Yet? FBA and BIP Technical Adequacy in Schools.” The authors should be commended for adding to the scholarship on this topic in a meaningful way with a large sample of reviewed functional behavior assessments (FBA) and behavior intervention plans (BIP). Furthermore, this topic continues to be important for volumes of individuals with disabilities within the K-12 education system in the United States. As the authors again demonstrated, the quality of FBAs and BIPs continues to negate the efforts at ameliorating behaviors that interfere with learning and/or safety and developing skills that promote learning and functioning in schools. The authors were clear in their multiple purposes within the current study and their methods, provided results, and discussion aligned to these areas.

Although not a part of the TATE, the authors may have reported the role(s) of the individual(s) responsible for conducting the FBA and developing the BIP as part of their findings. The inadequacy of FBAs and BIPs aligns in large part to the absence or inadequacy of the expertise, training, and/or supervision of those individuals within the K-12 system who are responsible for undertaking these processes. For this reason, scholarship that documents this relationship further contributes to the literature and serves to promote future research (and ideally, policy) related to the skills and proficiencies required of those responsible for developing FBAs and BIPs. The authors should be commended for their important work and I support acceptance of this manuscript for publication. 
